# Loss of XBP1 Leads to Early-Onset Retinal Neurodegeneration in a Mouse Model of Type I Diabetes

**DOI:** 10.3390/jcm8060906

**Published:** 2019-06-25

**Authors:** Todd McLaughlin, Manhal Siddiqi, Joshua J. Wang, Sarah X. Zhang

**Affiliations:** 1Departments of Ophthalmology and Ross Eye Institute, University at Buffalo, Buffalo, NY 14203, USA; toddmcla@buffalo.edu (T.M.); manhalsi@buffalo.edu (M.S.); jianxinw@buffalo.edu (J.J.W.); 2SUNY Eye Institute, State University of New York, Buffalo, NY 14203, USA; 3Department of Biochemistry, State University of New York, Buffalo, NY 14203, USA

**Keywords:** diabetic retinopathy, X-box binding protein 1, neurodegeneration, synapses, photoreceptors

## Abstract

Retinal neuronal injury and degeneration is one of the primary manifestations of diabetic retinopathy, a leading cause of vision loss in working age adults. In pathological conditions, including diabetes and some physiological conditions such as aging, protein homeostasis can become disrupted, leading to endoplasmic reticulum (ER) stress. Severe or unmitigated ER stress can lead to cell death, which in retinal neurons results in irreversible loss of visual function. X-box binding protein 1 (XBP1) is a major transcription factor responsible for the adaptive unfolded protein response (UPR) to maintain protein homeostasis in cells undergoing ER stress. The purpose of this study is to determine the role of XBP1-mediated UPR in retinal neuronal survival and function in a mouse model of type 1 diabetes. Using a conditional retina-specific XBP1 knockout mouse line, we demonstrate that depletion of XBP1 in retinal neurons results in early onset retinal function decline, loss of retinal ganglion cells and photoreceptors, disrupted photoreceptor ribbon synapses, and Müller cell activation after induction of diabetes. Our findings suggest an important role of XBP1-mediated adaptive UPR in retinal neuronal survival and function in diabetes.

## 1. Introduction

Diabetic retinopathy (DR) is a common complication of diabetes that can lead to severe vision loss [1]. While the mechanism for the onset and progression of vision deterioration in diabetes, likely due to retinal neuronal and vascular damage, is complex and not fully understood, it has become clear that a loss of protein homeostasis in the endoplasmic reticulum (ER) of retinal cells is a contributing factor in this process [2,3]. In this study, we focus on neuronal cell loss, while recognizing the likely contributions of vasculopathy to vision loss in DR. The protein dyshomeostasis, manifested as an overabundance of unfolded or misfolded proteins in the ER, leads to ER stress. This, in turn, activates the unfolded protein response (UPR), an adaptive mechanism consisting of three arms of signaling pathways mediated by inositol requiring enzyme 1 (IRE1), PKR-like endoplasmic reticulum kinase (PERK), and activating transcription factor 6 (ATF6), to attempt to restore cellular homeostasis [4]. Failure of the UPR to relieve ER stress often leads to cellular structural and functional disruption and even cell death. Among the three UPR branches, the IRE1 pathway is the most conserved signaling to combat protein dyshomeostasis through inducing the activation of a transcription factor, namely X-box binding protein 1 (XBP1) [5]. Given that cell injury and cell death of retinal neurons results in irreversible vision loss, the ability of XBP1 to mitigate ER stress in retinal cells under chronic disease conditions such as diabetic retinopathy is, therefore, of significant interest.

In a recent study, we examined the role of XBP1 in the development and maintenance of retinal neurons using a conditional knockout (cKO) mouse line, wherein XBP1 is deleted in retinal progenitor cells, generated by crossing XBP1 floxed mice with a Chx10 cre line [6]. Surprisingly, we found that XBP1 deletion does not affect retinal neuronal development; however, the loss of XBP1 in the retina leads to an early onset of age-related decline in retinal function, as indicated by reduced a-wave and b-wave amplitudes of electroretinogram (ERG) tests in 12–14 month-old mice [6]. In agreement, we found that the thicknesses of the retina and some individual retinal layers, including outer nuclear layer (ONL) and inner plexiform layer (IPL), and the number of retinal ganglion cells (RGC) are significantly reduced. Furthermore, we observed significantly increased abnormal ectopic neuronal synapses in the ONL and disrupted, discontinuous lamina synapses in the IPL [6]. These changes indicate that XBP1 could serve as a master regulator of the cellular adaptive system to chronic stresses during aging, and the deficiency of XBP1 compromises the ability of retinal neurons to cope with stress, and, thus, leads to retinal neurodegeneration.

Compelling evidence suggest that aging has a significant impact on diabetes and diabetic complications, not only that the unprecedented aging of the world’s population has become a major contributor to the diabetes epidemic, but also the incidence of macrovascular and microvascular complications appears to be the highest in older diabetic individuals [7,8,9,10]. The United Kingdom Prospective Diabetic Study suggests that older age (>58 years) is a risk factor for progression of DR [11]. Moreover, neuronal degeneration in DR, manifested by reduced retinal function, apoptosis and cell death of retinal neurons, disrupted function and integrity of retinal synapses, was also observed in the retina with aging [6,12,13]. These similarities may indicate common underlying mechanisms in age-related and diabetic retinal neurodegeneration. We hypothesize that XBP1, which functions as a key regulator of retinal neuronal adaptive mechanism to aging-related stresses, may also play an important role in maintaining retinal neuron function and survival in diabetes. In the present study, we investigated how the loss of XBP1 affects the function and morphology of the retina in a mouse model of type 1 diabetes.

## 2. Experimental Section

### 2.1. Animals

Generation of conditional knockout mice with XBP1 gene deletion in the retina (XBP1 cKO) was achieved by crossing XBP1 floxed mice carrying LoxP sites flanking exon 2 of the XBP1 gene [14] with a retina-specific Chx10-Cre line [15] as described previously [6]. Genotyping was performed by PCR with a primer set of three allele-specific primers for XBP1 WT, floxed, and exon 2-deleted alleles [6] or primers for Cre (forward: 5′-GCATTACCGGTCGATGCAACGAGTGATG-3′ and reverse: 5′-GAGTGAACGAACCTGGTCGAAATCAGTG-3′) with a Cre-positive animal resulting in a 408 bp band. Mice aged 8–9 weeks were randomly assigned to receive five consecutive daily intraperitoneal injections of streptozotocin (50 mg/kg body weight) in 0.1 M citrate buffer, or equivalent volume control citrate buffer. Blood glucose was determined with a glucometer and test strips (ReliOn Prime). Animals with persistent blood glucose levels greater than 14 mmol/L were considered diabetic. All animal procedures were approved by the Institutional Animal Care and Use Committees at the University at Buffalo, State University of New York, and in accordance with the guidelines of the Association for Research in Vision and Ophthalmology statements for the “Use of Animals in Ophthalmic and Vision Research”.

### 2.2. Electroretinography (ERG)

Dark-and light-adapted electroretinogram (ERG) were performed with a Diagnosys Espion ColorDome system and manufacturer installed software (Diagnosys LLC, Lowell, MA, USA) as previously described [6]. Immediately prior to testing, mice were anesthetized with intraperitoneal injection of 140 mg/kg ketamine and 7 mg/kg xylazine. After mice became nonresponsive, both pupils were dilated with 1% atropine (Falcon Pharmaceuticals) followed by 2.5% Phenylephrine Hydrochloride (Bausch & Lomb). An electrode was inserted into the tail as ground, and a reference electrode placed subcutaneously centrally between the eyes. An ophthalmic gel (Gonak, Akorn) was applied to each cornea immediately prior to placement of electrodes across each cornea. For light-adapted ERGs, animals were light adapted for at least 10 min prior to stimulation by five flashes of 4 ms duration at 1 Hz at 10 cd s/m^2^ with a background of 5 cd s/m^2^. The amplitude of the a-wave was recorded as the lowest point of the initial response compared to baseline. The amplitude of the b-wave was calculated from the a-wave peak.

Dark-adapted step ERGs were performed using a custom protocol designed within the Diagnosys software as previously described [6]. Briefly, mice were dark adapted overnight (approximately 16 h) prior to anesthetization and dilation as described above. A protocol consisting of ten series of three identical flashes of light of 4 ms duration was applied with a delay between each series of flashes of 15–60 s (the delay increases with flash intensity). Light intensity for each series from dimmest to brightest were (luminance in log (cd s/m^2^)): −3.6, −3.0, −2.4, −1.8, −1.2, −0.6, 0.0, 0.6, 1.4, 2.1. The amplitude for the a-wave was the lowest point of the initial response and the b-wave amplitude was measured from the a-wave peak and determined to be the peak after the oscillatory potentials and within 150 ms of the a-wave peak. The *n* value represents independent animals.

### 2.3. Retinal Immunohistochemistry and Morphometry

Enucleated eyes from freshly sacrificed mice were punctured in the cornea and immersion fixed in 4% w/v paraformaldehyde in phosphate buffered saline (PBS), pH 7.4 for one hour and washed in PBS multiple times. Prior to embedding in OCT, eyes were equilibrated in 30% sucrose w/v in PBS at 4C. Eyes were embedded in OCT filled molds and frozen in absolute ethanol chilled on dry ice. Cryosections were cut at 20 µm and mounted on Superfrost Plus slides (Statlab). Sections were blocked with PBS plus 0.5% Triton X-100 (PBSt) plus 1% BSA fraction V (Calbiochem) for approximately 1 h at room temperature (RT) and incubated with primary antibody overnight at 4C in a light-protected, humidified chamber. Sections washed with PBSt 3–5 times over 20–60 min, and incubated with the appropriate secondary antibody in PBSt plus 1% BSA for 1 h at RT in a light-protected, humidified chamber. Secondary antibody was washed 3–5 in PBSt and sections overlaid with Vectashield mounting medium with DAPI (Vector, H-1200) for examination. All antibodies used are listed in Table 1. Images were taken with an upright Olympus BX53 microscope using 10–40 × objectives and an Olympus DP80 digital camera. Confocal images were taken with a Leica SP8 confocal microscope as a z-stack through the entire thickness of the section and analyzed in Image J Fiji as a maximum z-projection. Images were processed, montaged, and analyzed with Adobe Photoshop and Image J Fiji.

### 2.4. Image Analyses

All analyses were performed by two independent authors blinded to genotype and glycemic condition. Brn3a-positive cells in the GCL were counted across the entire retina in central sections using a custom macro in Image J as previously described [6]. For all other measurements, images taken within subdomains between 400–1000 µm of the optic disk were used. Measurements of retinal layer thickness were done as previously described [6]. Ribbon synapses were quantified blind to genotype and condition using the Find Maxima function in Image J. All images were prepared and photographed identically and were thresholded with identical Noise Tolerance. Preliminary quantification was tested against manual counts prior to subjecting experimental cases to the procedure. GFAP-positive processes were counted blind to condition and genotype by two independent observers. Labeled processes extending across the entire INL were counted. Cone cells were counted using z-projections of z-stacks obtained on a confocal microscope using identical settings across all cases. For each analysis, 3–5 images were used per animal.

### 2.5. Statistical Analyses

One-way ANOVA with Bonferroni post-hoc test was used for multiple group comparisons and Student’s *t*-test was used for pairwise comparisons. Significance was determined as *p* < 0.05.

## 3. Results

### 3.1. Conditional Deletion of XBP1 in the Retina Does Not Alter Retinal Function or Morphology at the Onset of Diabetes

Previously, we demonstrated that there is no significant difference in retinal function and morphology between XBP1 cKO (XBP1 fl/fl; Chx10-cre) mice and wild type (WT; XBP1 fl/fl) mice through 8 months of age [6]. Herein, we induced diabetes with five consecutive daily injections of STZ in XBP1 cKO and WT mice at 8–9 weeks of age. To validate our previous finding that XBP1 deletion does not affect retinal development and to exclude any potential effect resulting from the STZ injection, we performed an initial functional study to uncover any differences between WT and XBP1 cKO mice at 2 weeks after STZ injection. We found no significant differences in retinal function between diabetic and non-diabetic WT and XBP1 cKO mice. Dark-adapted (scotopic) electroretinogram (ERG) for WT and XBP1 cKO mice showed identical a-wave amplitude and b-wave amplitude in response to each of a series of light flashes of increasing intensity (Figure 1A). Similarly, light-adapted (photopic) transient ERGs demonstrated no significant difference between WT and XBP1 cKO mice, as both a-wave and b-wave responses are indistinguishable (Figure 1B). Thus, the retina of XBP1 cKO mice appears and functions identically to WT at the time of onset of diabetes.

### 3.2. Conditional Deletion of XBP1 Results in Early Decline of Retinal Function in Diabetic Mice

There was no significant difference in blood glucose and body weight between diabetic WT and diabetic XBP1 cKO mice with 20 weeks of hyperglycemia (Table 2). Examination of retinal function revealed no significant differences in a-wave or b-wave responses for either scotopic or photopic ERG in diabetic WT mice compared to non-diabetic WT controls at this stage (Figure 2). Similarly, age-matched non-diabetic XBP1 cKO mice were indistinguishable in both scotopic and photopic ERG responses from non-diabetic WT mice (Figure 2). In contrast, we found significant deficits in the ERGs of diabetic XBP1 cKO mice compared to the other groups. Scotopic ERGs revealed a significant decline in b-wave amplitude in diabetic XBP1 cKO mice, especially at higher light intensities (Figure 2A). Similarly, photopic ERGs revealed a significant decline in the b-wave amplitude of diabetic XBP1 cKO mice compared to every other group (Figure 2B). Interestingly, we found no significant differences in a-wave amplitude in either scotopic or photopic ERGs for any group (Figure 2).

### 3.3. Conditional Deletion of XBP1 Leads to Retinal Degeneration in Diabetic Mice after 20 Weeks of Hyperglycemia

After 20 weeks of hyperglycemia, retinal morphology was examined in diabetic and age-matched non-diabetic XBP1 cKO mice and WT mice. Quantification of total retinal thickness and that of individual retinal layers revealed an overall thinning of the retina in both diabetic WT and XBP1 cKO mice (Figure 3). Specifically, diabetic WT retinas were significantly thinner than non-diabetic WT retinas at this stage and diabetic XBP1 cKO retinas were thinner than all other groups (Figure 3B). The largest decline of retinal layer thickness in diabetic XBP1 cKO retinas was in the ONL, with a smaller decrease in the INL and no significant difference in the IPL (Figure 3C). Each of the ONL, INL, and IPL was slightly thinner in diabetic WT retina compared to non-diabetic WT control; however, no individual layer measurement reached significance (Figure 3C). Thus, after 20 weeks of hyperglycemia, diabetic XBP1 cKO mice showed retinal degeneration to a greater extent than diabetic WT mice, when compared to age-matched non-diabetic mice.

### 3.4. Loss of Rod and Cone Photoreceptors and Outer Segment Disorganization in Diabetic XBP1 cKO Retinas

The most prominent change in retinal layer thicknesses of diabetic XBP1 cKO mice occurred in the ONL. Further examination of the ONL, comprised of photoreceptor cell bodies, revealed a loss of both rods and cones in diabetic XBP1 cKO mice. We used an anti-cone arrestin antibody to label and quantify cone photoreceptors in the ONL (Figure 4). Cone arrestin labels cone cells and their processes, though the cone cell bodies are localized to the outer aspect of the ONL (Figure 4A). We found a significant loss of cone photoreceptors in diabetic XBP1 cKO retinas compared to every other group (Figure 4B). There was a small decline in cone cells in the retina of diabetic WT mice compared to non-diabetic retinas, but this did not reach significance. Similarly, the ONL in diabetic XBP1 cKO retinas had fewer DAPI-labeled nuclei, indicating an overall loss of photoreceptors (Figure 4C). Once again, diabetic WT retina had an approximate 10% reduction in cells in the ONL, but the decrease did not reach significance when compared to non-diabetic retina (Figure 4C). Though we quantified only cone cells directly, our data suggest that there is also a loss of rod photoreceptors in diabetic XBP1 cKO retina because the vast majority of cells in the ONL are rod photoreceptors [16].

Cone arrestin also labels cone processes, including the inner and outer segments. We noticed a striking array of cone arrestin fragments external to the photoreceptor outer segments (OS) (Figure 4A). We found that diabetic WT retinas have significantly more cone arrestin positive fragments in the OS than non-diabetic WT retinas (Figure 4D). This fragmentation was more severe in diabetic XBP1 cKO retinas, which had significantly more cone arrestin-positive fragments than every other group (Figure 4D). Thus, we conclude that diabetic XBP1 cKO retina experience a greater loss of photoreceptors and a greater apparent fragmentation of cone outer segments than age-matched non-diabetic mice or diabetic WT mice.

### 3.5. Loss of RGCs in Diabetic XBP1 cKO Mice

In addition to cell loss in the ONL, we find a small, yet significant depletion of RGCs specifically in diabetic XBP1 cKO retinas. We used the RGC marker, Brn3a, to label the majority of RGCs in the ganglion cell layer (GCL; Figure 5A). There was no difference in the number of Brn3a-positive cells between non-diabetic WT, non-diabetic XBP1 cKO, or diabetic WT mice (Figure 5B). However, in diabetic XBP1 cKO retinas, we found a significant approximately 12% decrease in Brn3a-positive RGCs in the GCL of diabetic XBP1 cKO mice after 20 weeks of hyperglycemia (Figure 5B). Thus, diabetic XBP1 cKO mice experience a loss of RGCs at an earlier stage of diabetic retinopathy than WT mice.

### 3.6. Fewer Ribbon Synapses in the ONL of Diabetic XBP1 cKO Mice

The functional deficits in diabetic XBP1 cKO mice revealed by ERG led us to examine the outer plexiform layer (OPL), where synapses between photoreceptors and bipolar cells reside. Using an anti-Ribeye antibody to label ribbon synapses between photoreceptors and bipolar cells, we quantified the number of prominent ribbon synapses in each group (Figure 6A). We found no difference in the number of ribbon synapses in non-diabetic XBP1 cKO mice compared to non-diabetic WT mice (Figure 6B). However, we found a trend for approximately 12% fewer ribbon synapses in diabetic WT mice compared to non-diabetic WT and XBP1 cKO mice, though this did not reach significance. In contrast, diabetic XBP1 cKO mice had approximately one-third fewer Ribeye-positive ribbon synapses than non-diabetic WT and XBP1 cKO, and approximately 25% fewer than in diabetic WT retina (Figure 6B). Thus, though we saw some degeneration in diabetic WT compared to non-diabetic mice, we found a greater extent of degeneration in diabetic XBP1 cKO mice.

### 3.7. GFAP Elevation in Müller Cells

In addition to morphological measurements, we examined the level of glial fibrillary acidic protein (GFAP), a marker for glial activation associated with neuronal injury and diabetes, in Müller cells (Figure 7A; [17]). GFAP-positive processes extending across the INL are rare in non-diabetic WT mice. In non-diabetic XBP1 cKO, diabetic WT, and diabetic XBP1 cKO mice we found multiple GFAP-positive Müller cell processes extending across the INL and extending into the OPL (Figure 7). There was a significant increase in the number of GFAP positive Müller cell processes in diabetic XBP1 cKO retinas compared to diabetic WT retinas and non-diabetic retinas (Figure 7B). Interestingly, we found an upregulation of GFAP on Müller cell processes in non-diabetic XBP1 cKO and diabetic WT retinas compared to the normal low-level expression in non-diabetic WT retina, but this was not statistically significant.

## 4. Discussion

In the present study, we investigated the role of XBP1 in retinal function and structural integrity in experimental diabetes. Our data suggest that a lack of XBP1 leads to an early decline of retinal function, accompanied by loss of retinal photoreceptors and RGCs and disruption of retinal synapses, in diabetic mice. Importantly, deletion of XBP1 in Chx10-cre-expressing retinal progenitor cells, which give rise to most retinal neurons and a subset of Müller cells [15], does not affect retinal function or structure prior to the onset of diabetes nor affects blood glucose or body weight in diabetic mice. In contrast, diabetic XBP1 cKO mice, after a 20-week hyperglycemia, demonstrate multiple deficits in retinal function and structure that are not found in diabetic WT mice and display more severe neuronal damage compared to diabetic WT mice. These changes include a loss of RGCs, cone photoreceptors, and rod photoreceptors, as well as a reduction of the number of Ribeye-positive ribbon synapses in the OPL. Functionally, we find a decline in the b-wave for both dark and light-adapted ERGs specifically in diabetic XBP1 cKO mice but not in diabetic WT mice. These findings extend our previous observations in aging retina [6], supporting the notion that XBP1 plays a pivotal role in maintaining retinal neuron survival and function under chronic stress conditions including aging and diabetes.

One major finding from the current study is that deletion of XBP1 in the retina accelerates retinal function deterioration in diabetes. Notably, we did not observe any significant alteration in retinal function, measured by ERG, in diabetic WT mice after 20 weeks of hyperglycemia compared to non-diabetic controls. This finding is, in part, in line with previous reports [18,19,20]. In most of these studies, abnormal changes in ERG a-waves and/or b-waves were observed in diabetic mice after 6–12 months of hyperglycemia, with considerable variation in the timing and scope of the changes. More sensitive measures, such as scotopic threshold response (STR) and oscillatory potentials (OPs), that are believed to represent direct measures of inner retinal function derived from RGCs, bipolar cells, and amacrine cells, have been shown to be able to detect earlier functional deficits in diabetic retinas [1,21]. In addition, the optokinetic response, which determines spatial frequency threshold and contrast sensitivity, also enables the detection of early subtle visual defects in diabetic animals, e.g., in 5-month-old diabetic Akita mice [22]. Thus, future studies taking advantage of these new measures are warranted to evaluate whether the inner retinal function is more severely affected, corresponding to the loss of RGCs and cells in the INL, in diabetic XBP1 cKO mice compared to diabetic WT mice and non-diabetic controls. Similarly, multiple morphometric analyses in diabetic Akita mice, including number of synaptic ribbons in the OPL and number of cone photoreceptors, reveal no significant differences in 6-month-old mice with defects present in 9-month-old mice [23]. These deficits are of a similar scale to those we report here after just 20 weeks of hyperglycemia in diabetic XBP1 cKO mice.

Because the standard ERG is a composite signal derived from the combination of cell populations across the entire retina [1], it is difficult to identify the exact mechanistic cause of the ERG deficits in diabetic XBP1 cKO mice. We speculate that the significant loss of cone and rod photoreceptors is a likely contributing factor to the b-wave decline in light-adapted and dark-adapted ERG, respectively, though interestingly we found no changes in a-wave amplitudes, driven by photoreceptors, even with the substantial loss of photoreceptors in diabetic XBP1 cKO mice. Additionally, the reduction in ribbon synapses may contribute to both light-and dark-adapted ERG dysfunction in diabetic XBP1 cKO mice. A recent report also indicates a significant reduction of bipolar cell dendrite length in diabetic Akita mice at 9-months old when ERG responses are significantly affected, but not at 6-months-old when only subtle ERG changes were reported [23]. Thus, the malformation, or the loss of structural integrity, of the ribbon synapses could be another potential contributing factor to b-wave decreases in diabetic XBP1 cKO mice. In addition, it was found that the content of synaptic proteins, including the presynaptic vesicle protein synaptophysin, was significantly reduced in diabetic retinas [24], likely due to alternations in synaptic protein synthesis, post-translational modification, and turnover [25]. These changes, in turn, compromise synaptic function, attenuate neurotransmission, and subsequently affect visual signal processing, resulting in functional deterioration prior to structural damage of the retina in DR. Collectively, these data suggest an important role of protein dysregulation and dyshomeostasis in the pathogenesis of retinal neurodegeneration and visual impairment in diabetes.

XBP1 is a major transcription factor in the core pathway of the UPR that regulates protein homeostasis in cells undergoing ER stress [2]. Previous studies found that activation of IRE1/XBP1 and PERK/ATF4/CHOP pathways differentially regulate cell death, inflammation, and vascular permeability in the retina [2,26,27,28,29,30,31]. Preconditioning with mild ER stress activates XBP1-dependent UPR, inhibiting endothelial inflammation and reduces vascular leakage [32]. Loss of XBP1 induces Müller glia activation and promotes retinal inflammation in DR [33]. In cultured 661W cells, downregulation of XBP1 increases ER stress-induced inflammatory cytokine CXCL10 and CCL2 expression, suggesting a potential role of XBP1 in regulation of photoreceptor inflammation [31]. Moreover, XBP1-deficient RPE cells are susceptible to oxidative stress-induced apoptosis and cell death and tight junction damage [34,35,36,37]. These findings imply a vital role of XBP1 in maintaining retinal function and integrity during stress conditions. Interestingly, a recent proteomic study shows that the UPR pathways are uniquely enriched in control retinas from healthy donors but not from the diabetics [38]. Furthermore, XBP1 activation was found significantly reduced in the retina with aging. These results suggest insufficient XBP1 function could contribute to retinal cell injury and neurodegeneration associated with aging and diabetes. Our current study tested this hypothesis and our data support a protective role of XBP1 against retinal neuronal injury and dysfunction induced by diabetes. The exact mechanisms by which XBP1 protects retinal neurons in DR remain to be investigated in future studies.

In addition to neuronal phenotypes, a characteristic of DR, and neuronal injury in general, is retinal glial activation [39]. We examined Müller cell activation by determining the levels of GFAP in Müller cell processes, a hallmark of glial activation. Müller cells are the primary retinal glia and involved in regulation of blood flow, neurotransmitter uptake, and metabolism, so injury or dysfunction in Müller cells would introduce multiple potential pathways to contribute to neurodegeneration and functional decline [40]. In diabetes, Müller cell secreted VEGF promotes retinal inflammation and increases vascular permeability [41]; however, reducing Müller cell density by depletion of VEGFR2 expression in Müller glia accelerates retinal neuron degeneration in DR, in part due to decreased production of neurotrophic factors from Müller cells [42]. In the current study, we observed significantly increased Müller cell activation in diabetic retinas. We speculate that increased activation of Müller cells in diabetic WT mice, and to a greater extent in diabetic XBP1 cKO mice, is secondary to neuronal injury and synaptic damage. While Chx10-Cre is expressed primarily in developing neurons and mature bipolar cells, weak expression of Chx10-cre was observed in Müller cells [43]. Whether deletion of XBP1 affects Müller cell-derived neurotrophic factor production, which, in turn, augments retinal neuron degeneration in XBP1 cKO mice remains elusive.

In summary, we have shown that deletion of XBP1 in retinal neurons, and likely some Müller cells, results in accelerated and aggravated neural retinal dysfunction and degeneration, accompanied by enhanced Müller glia activation in diabetes. The molecular mechanisms behind the retinal degeneration in XBP1 cKO mice, possibly including dysregulation of synaptic proteins and synapse formation, increased neuronal cell death, and disruption of cellular metabolism, are currently being investigated.

## Figures and Tables

**Figure 1 jcm-08-00906-f001:**
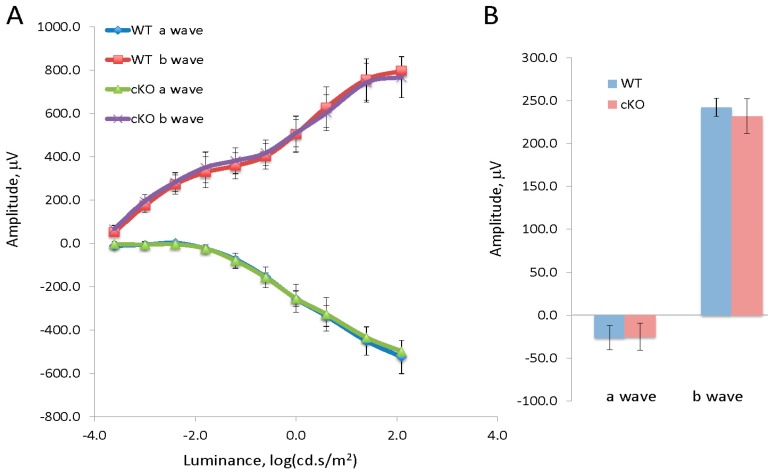
(**A**) Graph of the a-wave and b-wave responses of a dark adapted ERG with 10-steps of increasing stimulus luminance for wild type (XBP1 fl/fl; WT) mice (*n* = 8) and XBP1 fl/fl; Chx10-cre positive (cKO) mice (*n* = 12); (**B**) Graph of the a-wave and b-wave responses of a transient, light-adapted ERG for WT (*n* = 6) and XBP1 fl/fl; Chx10-cre positive mice (*n* = 12). Data are expressed as mean ± S.D.

**Figure 2 jcm-08-00906-f002:**
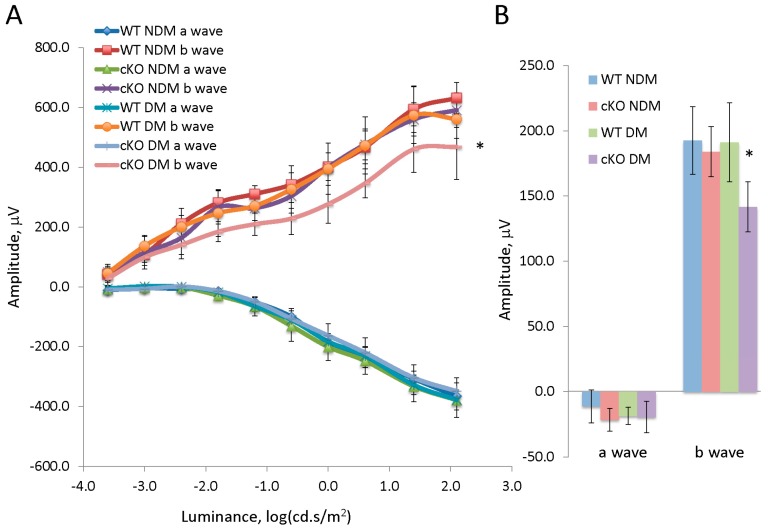
(**A**) Graph of the a-wave and b-wave responses of a dark adapted ERG with 10-steps of increasing stimulus luminance for non-diabetic (NDM) and 20-week diabetic (DM) wild type (WT; NDM *n* = 5, DM *n* = 4) and XBP1 fl/fl; Chx10-cre positive (cKO; NDM *n* = 4, DM *n* = 7) mice; (**B**) Graph of the a-wave and b-wave responses of a transient, light-adapted ERG for non-diabetic (NDM) and diabetic (DM) wild type (WT; NDM *n* = 4, DM *n* = 5) and XBP1 fl/fl; Chx10-cre positive (cKO; NDM *n* = 5, DM *n* = 5) mice. Mean ± S.D.; * *p* < 0.05.

**Figure 3 jcm-08-00906-f003:**
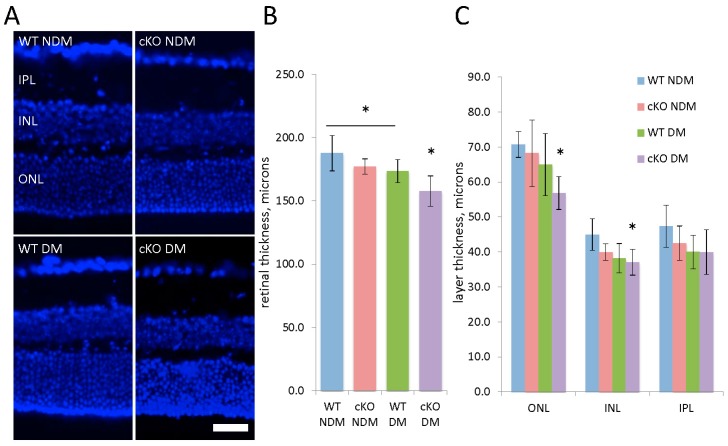
(**A**) Retinal cryosections (20 µm) stained with DAPI (blue) to label nuclei and identify retinal layers for wild type (WT) non-diabetic (NDM; *n* = 5) and 20-week diabetic (DM; *n* = 5) mice and XBP1 fl/fl; Chx10-cre positive (cKO) NDM (*n* = 5) and DM (*n* = 6). Scale bar = 40 µm; (**B**) Graph of total retinal thickness indicates the WT DM retina is significantly thinner than WT NDM retina and the cKO DM retina is thinner than all other groups; (**C**) Graph depicting the thickness of the outer nuclear layer (ONL), inner nuclear layer (INL), and inner plexiform layer (IPL) shows some retinal thinning across all layers with significant decreases in XBP1 cKO DM ONL and INL. Mean ± S.D.; * *p* < 0.05.

**Figure 4 jcm-08-00906-f004:**
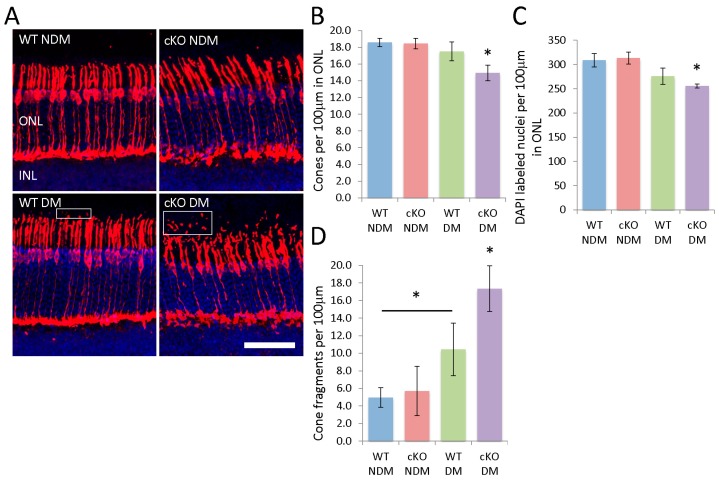
(**A**) Retinal cryosections (20 µm) labeled with anti-cone arrestin to identify cone photoreceptors and counterstained with DAPI (blue) to label nuclei. Photoreceptor fragments exterior to the outer segments also label positively for cone-arrestin (boxes). Scale bar = 50 µm; Graphs depict the number of (**B**) cone arrestin positive cells and (**C**) DAPI-positive nuclei per 100 µm of the ONL. XBP1 cKO mice have significantly fewer cone photoreceptors and DAPI labeled nuclei; (**D**) Graph depicts the number of cone arrestin labeled fragments exterior to the outer segments for each group. There is a significant increase in cone arrestin positive fragments in DM mice, with a greater increase in XBP1 cKO DM mice. *n* = 3 for all groups; Mean ± S.D.; * *p* < 0.05.

**Figure 5 jcm-08-00906-f005:**
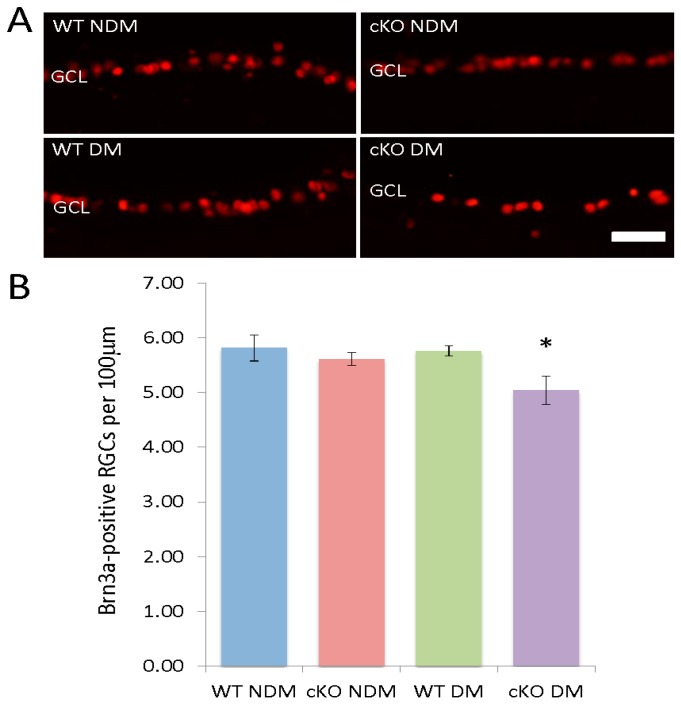
(**A**) Retinal cryosections (20 µm) labeled with anti-Brn3a (red) to identify retinal ganglion cells in the ganglion cell layer (GCL) of non-diabetic (NDM) and 20-week diabetic (DM) wild type (WT; NDM *n* = 4, DM *n* = 4) and XBP1 fl/fl; Chx10-cre positive (cKO; NDM *n* = 5, DM *n* = 6) mice; (**B**) Graph depicts the number of Brn3a-positive cells per 100 µm in the GCL for each group. Mean ± S.D.; scale bar = 50 µm; * *p* < 0.05.

**Figure 6 jcm-08-00906-f006:**
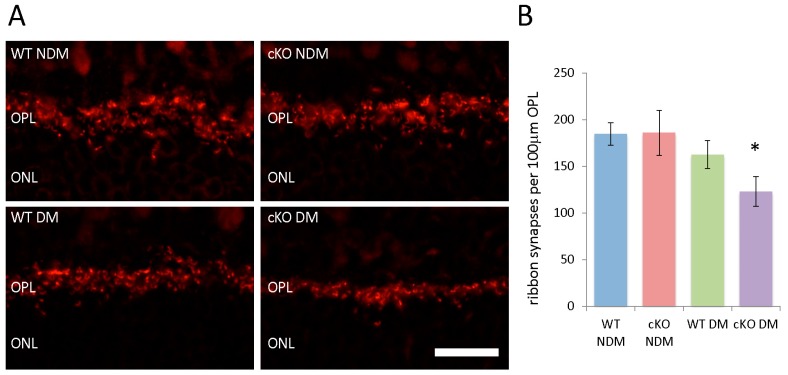
(**A**) Retinal cryosections (20 µm) labeled with antibodies against Ribeye (red), a synaptic marker for ribbon synapses in the outer plexiform layer (OPL), in non-diabetic (NDM) and 20-week diabetic (DM) wild type (WT) and XBP1 fl/fl; Chx10-cre positive (cKO) mice; (**B**) Graph depicts the average number of Ribeye-positive ribbon synapses per 100 µm of the OPL for each group. There is a significant decrease in the number of ribbon synapses for XBP1 cKO DM mice. *n* = 3 for all groups; Mean ± S.D.; scale bar = 50 µm; * *p* < 0.05. ONL, outer nuclear layer.

**Figure 7 jcm-08-00906-f007:**
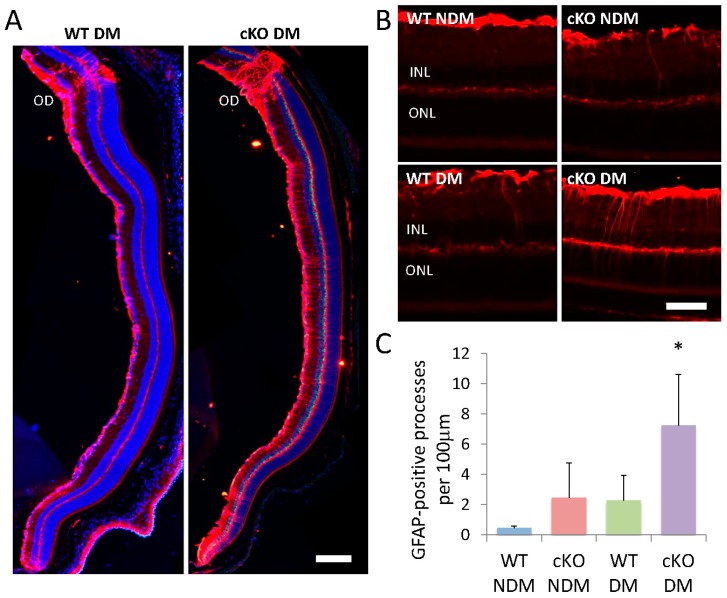
(**A**,**B**) Retinal cryosections (20 µm) labeled with antibodies against GFAP (glial fibrillary acidic protein; red) of non-diabetic (NDM) and 20-week diabetic (DM) wild type (WT; NDM *n* = 3, DM *n* = 4) and XBP1 fl/fl; Chx10-cre positive (cKO; NDM *n* = 4, DM *n* = 5) mice; (**C**) Graph depicts the average number of GFAP-positive processes per 100 µm in the inner nuclear layer (INL) for each group. Mean ± S.D.; scale bar in A = 150 µm, scale bar in B = 50 µm; * *p* < 0.05. OD, optic disk; ONL, outer nuclear layer.

**Table 1 jcm-08-00906-t001:** Antibodies used in immunofluorescence.

Antibody	Dilutions	Catalog No.	Company
anti-Ribeye, B-domain	1:800	192 003	Synaptic Systems
anti-Cone Arrestin	1:1000	AB15282	Millipore
anti-Brn3a	1:800	sc-31984	Santa Cruz Biotechnology
anti-GFAP	1:800	Z0334	Dako
Alexa Fluor-488	1:800	A11001	Molecular Probes
Alexa Fluor-594	1:800	A11005	Molecular Probes
Alexa Fluor-594	1:800	A11080	Molecular Probes
Texas Red	1:800	T6391	Molecular Probes

**Table 2 jcm-08-00906-t002:** Animal body weight and blood glucose level.

		Body Weight	Blood Glucose
	*n*	(g)	(mmol/L)
WT NDM	8	26.6 ± 1.5	6.6 ± 0.8
cKO NDM	8	25.9 ± 1.1	6.5 ± 0.6
WT DM	8	21.0 ± 1.3	23.3 ± 2.1
cKO DM	8	20.8 ± 1.8	24.1 ± 2.5
*p* values		*p* < 0.01	*p* < 0.01

Data are shown as mean ± S.D.; *p* < 0.01 for DM vs NDM. DM: diabetic; NDM: non-diabetic.

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
