# Peer review of "Loss of XBP1 Leads to Early-Onset Retinal Neurodegeneration in a Mouse Model of Type I Diabetes"

_jcm, 2019, doi:10.3390/jcm8060906_

Reviewer 1 Report

Comments to the authors

The manuscript by McLaughlin et al. reports interesting findings of the protective function of XBP1 against retinal neuronal injury and dysfunction induced by DM. However, the mechanism of XBP1 protection against ER stress in this model is missing. ER stress should be confirmed in the model.

Major comments:

1) Vision loss is not only caused by neuronal stress or death, but also by tractional retinal detachment due to fibrotic responses in the retinal tissue. Vision loss is mostly occurring at the advanced stage of DR, PDR. The authors should place their findings within the DR field, where both neuropathy and vasculopathy are contributors to visual impairment, and where whether neuropathy or vasculopathy is the primary event is still not well understood.

2) The mechanism of XBP1 protection against ER stress in this model is missing. ER stress should be confirmed in the model.

Minor comments:

1) Colour of the bars should be made consistent throughout the Figures (e.g. Figure 3B and C).

2) Alignment of graphs in Figure 4B and D

3) Cone arrestin positive fragments should be indicated in Figure 4A also in the diabetic WT panel.

Author Response

Reviewer 1

The manuscript by McLaughlin et al. reports interesting findings of the protective function of XBP1 against retinal neuronal injury and dysfunction induced by DM. However, the mechanism of XBP1 protection against ER stress in this model is missing. ER stress should be confirmed in the model.

Major comments:

1)    Vision loss is not only caused by neuronal stress or death, but also by tractional retinal detachment due to fibrotic responses in the retinal tissue. Vision loss is mostly occurring at the advanced stage of DR, PDR. The authors should place their findings within the DR field, where both neuropathy and vasculopathy are contributors to visual impairment, and where whether neuropathy or vasculopathy is the primary event is still not well understood.

Response: We totally agree with the reviewer that both neuropathy and vasculopathy contribute to vision loss in DR. We have modified the wording in the introduction to reflect the changes (please see Lines 30 – 34). The focus of this current work is on the neuropathic elements as our conditional Xbp1 deletion occurs in retinal neurons.

2)    The mechanism of XBP1 protection against ER stress in this model is missing. ER stress should be confirmed in the model.

Response: This an excellent suggestion. ER stress can be indicated by the activation of the unfolded protein response (UPR), which is measured by the increase in GRP78 level, phosphorylation of IRE1α and PERK, and increased cleavage of ATF6 protein. Our data using quantitative RT-PCR revealed no significant change in the expression of ER stress response genes, including GRP78, ATF6, ATF4, and CHOP (the latter two are downstream effectors of PERK pathway), in this model. This data need to be confirmed at the protein level. However, we do not have any retinal samples available for protein analysis. Thus, based on our current data, we cannot conclude whether ER stress is altered in this model. We are now generating new diabetic mice to study the mechanisms underlying the retinal phenotype reported here. We will evaluate ER stress in the retina at multiple time points after diabetes onset in this model.

Minor comments:

1)    Colour of the bars should be made consistent throughout the Figures (e.g. Figure 3B and C).

Response: We have modified the figures as requested by using consistent colors throughout.

2)    Alignment of graphs in Figure 4B and D

Response: Graphs in 4B and 4D have been aligned.

3)    Cone arrestin positive fragments should be indicated in Figure 4A also in the diabetic WT panel.

Response: An inset box has been added to Figure 4A WT DM panel to highlight some cone arrestin positive fragments in that panel.

Reviewer 2 Report

In this manuscript, the authors investigated the role of XBP1 in type 1 diabetes-related early onset retinal neurodegeneration using a knockout mouse model. The manuscript was well written though there are a few concerns that require the authors' attention.

1) How many images were used for the immunohistochemical analysis? For instance, how many images were used for the counting of GFAP+ strands?

2) How did the authors ensure that for their immunohistochemical quantification, images from the same retinal regions were quantified?

3) The changes that the authors report in terms of retinal layer thickness as well as cell count in the ONL and RGC seem quite small. While the decrease appears to be statistically significant, could the authors comment on the biological significance in terms of their effect on overall function?

4) If the retinal thinking is due to loss of cell bodies, could the authors include a TUNEL assay or another equivalent test to reflect this?

Author Response

Reviewer 2

In this manuscript, the authors investigated the role of XBP1 in type 1 diabetes-related early onset retinal neurodegeneration using a knockout mouse model. The manuscript was well written though there are a few concerns that require the authors' attention.

1)    How many images were used for the immunohistochemical analysis? For instance, how many images were used for the counting of GFAP+ strands?

Response:  The n values reported indicate individual animals. For each immunohistochemical analysis 3-5 images per animal were examined. We have added these details to the manuscript (Lines 138 – 147).

2)    How did the authors ensure that for their immunohistochemical quantification, images from the same retinal regions were quantified? 

Response: We took extra care to ensure that the same areas of the retina were sampled for each analysis across cases. We used sections through or adjacent to the optic disk and 400mm -1000mm from the disk, as briefly described in Experimental Section (Lines 138 – 147). For Brn3a quantification we used the entire retinal section, as described (Line 137).

3)    The changes that the authors report in terms of retinal layer thickness as well as cell count in the ONL and RGC seem quite small. While the decrease appears to be statistically significant, could the authors comment on the biological significance in terms of their effect on overall function? 

Response: This is a valid question. We believe that the changes in retinal layer thickness only partially contribute to the functional deterioration and could be secondary to the dysfunction and damage of neuronal synapses. Though there is substantial variation in the literature, our findings are generally similar in the extent of loss observed in other studies (e.g., Hombrebueno et al., PlosOne 2014, at 9-months diabetic report similar cone loss and ribbon synapse loss). We have added some points to the Discussion regarding the timing for detection of declined function, neuronal loss, and synaptic changes in DR (Lines 334 – 338, and 346 - 349). Similarly, the extent of the loss of RGCs is similar to what we have previously observed in aging mice lacking Xbp1 (McLaughlin et al., Mol Neurodegen, 2018).

If the retinal thinking is due to loss of cell bodies, could the authors include a TUNEL assay or another equivalent test to reflect this?

             Response: We thank the reviewer for this great suggestion. We have used all sections for retinal morphometry, and immunolabeling with various markers for quantification of retinal neurons, synapses, and glial activation. We are currently generating a new cohort of diabetic mice for measuring metabolic changes in the cKO retinas. We will include TUNEL assay in this study.

Reviewer 3 Report

In the present study, McLaughlin et al. investigated how the loss of XBP1 affects the function and morphology of the retina in a mouse model of type 1 diabetes mellitus (DM).

The study is of potential interest, as the diabetic retinopathy (DR) –such as other DM complications- is bound to become widespread in the coming years, with a negative impact in economic terms for the developed countries. The paper is written in an appropriate language; generation of conditional knockout mice was correctly conducted; and in general experiments were adequately designed and performed, although not always the outer segments are visible in the retinal immunohistochemistries.

However, although the Authors do demonstrate their thesis , they  completely disregard the possibiltly they have in hnad  to unveil the underlying mechanism.

The manuscript is however severely incomplete in that the Authors totally ignored the rods in a mouse retina, which is not acceptable, as rods represent the vast majority of the retinal photoreceptors in the mouse retina, which  is rod-dominated. Cones constitute only 3% of the photoreceptors, distributed across the retina (see Carter-Dawson, and  LaVail, J. Comp. Neurol., 188 (1979), 245-62 and pp. 263-72).

The work could provide an advance towards the current knowledge on both DR, and AMD if it discussed the data in light of the new findings of a source of oxidative stress inside the mouse and bovine photoreceptor outer segments (see doi: 10.1111/bph.13173.)

 The mitochondrial redox chain proteins as well as ATP synthase were shown to be ectopically expressed, not only in the mitochondria. In particular, this was found in the rod Outer Segments (see the data from Calzia, et al.): in 2019 Authors cannot ignore the data on the existence of an extra-mitochondrial oxidative phosphorylation in the rod OS (see Bruschi et al doi: 10.1021/acs.jproteome.7b00741.). Notably, an extra-mitochondrial oxidative phosphorylation was reported also in many cellular membranes (see for example the work from Mangiullo, et al., doi: 10.1016/j.bbabio.2008.08.003.; Arakaki et al., Mol Cancer Res2003 Nov;1(13):931-9; Shimizu N, et al. Am J Physiol Heart Circ Physiol. 2007 Sep;293(3):H1646-53

Data from ERG show significant change only in the  a-wave of the electroretinogram,   which reflects the activity of photoreceptors, interstingly the only one  affected. Clearly the absence of effect on b-wave rules out a major role played by Mueller cells and inflamation, which imay instead be a consenquence of the oxidative stress primarily occurring in the rod OS due to hypèoxia and hyschemia due to impaiired blood supply, caused by the disease. . Undoubtedly,  the primary retinal damage occurs to photoreceptors, and rods in particular, and especially the outer segments (OS). In fact, oxidative stress is produced not only in the inner segment containing the mitochondria, but also in the Outer limb (see the cited work from Roelke et al. and Clazia et al, for example doi: 10.1016/j.biochi.2016.03.016.).

I recommend that Authors reformulate the paper taking into consideration these emerging data, Authors must perform retinal immunohistochemistry utilizing and antibody against one of the subunits of the proteins of the respiratory chain or to FoF1-ATP synthase, in order to follow the morphology of the rod OS, evidencing at least morphologically its early impairment. Nice would also be the ability to test one of the respiratory complexes on retinal unfixed sections, but this thechnique may be longer to perform 

Also,  the mouse model of AMD obtained by stress induced by Blue light irradiation (Roelke et al. doi: (10.1371/journal.pone.0071570) must be cited.

Author Response

Reviewer 3

In the present study, McLaughlin et al. investigated how the loss of XBP1 affects the function and morphology of the retina in a mouse model of type 1 diabetes mellitus (DM).

The study is of potential interest, as the diabetic retinopathy (DR) –such as other DM complications- is bound to become widespread in the coming years, with a negative impact in economic terms for the developed countries. The paper is written in an appropriate language; generation of conditional knockout mice was correctly conducted; and in general experiments were adequately designed and performed, although not always the outer segments are visible in the retinal immunohistochemistries.

Response: Many thanks for your encouraging and positive comments. Regarding immunohistochemistry, we took measures (described in Experimental Section, (Lines 137 – 147) to ensure equal sampling for the cone arrestin labeling and found the appearance of outer segments across cases to be consistent within groups.

However, although the Authors do demonstrate their thesis , they  completely disregard the possibiltly they have in hnad  to unveil the underlying mechanism.

Response: As discussed in the manuscript, the major purpose of this manuscript is to report the effect of XBP1 deletion in retinal neurons on retinal structure and function. XBP1 is transcription factor that regulates cellular response to ER stress. We found (using quantitative RT-PCR) that some ER stress markers are not altered in XBP1-deficient retinas. We are currently generating new cohort of diabetic mice to study the mechanisms underlying the exacerbated retinal damage in XBP1 cKO mice.

The manuscript is however severely incomplete in that the Authors totally ignored the rods in a mouse retina, which is not acceptable, as rods represent the vast majority of the retinal photoreceptors in the mouse retina, which  is rod-dominated. Cones constitute only 3% of the photoreceptors, distributed across the retina (see Carter-Dawson, and  LaVail, J. Comp. Neurol., 188 (1979), 245-62 and pp. 263-72).

Response: We agree with the reviewer that rods constitute vast majority of retinal photoreceptors. Thus, we believe that the reduction in ONL thickness (Figure 3C) largely represent the changes in rods. We have also quantified the rows of nuclei in ONL, which further confirms loss of rods in XBP1 cKO retina. These data are presented in Figure 4C and are described in the results section.

The work could provide an advance towards the current knowledge on both DR, and AMD if it discussed the data in light of the new findings of a source of oxidative stress inside the mouse and bovine photoreceptor outer segments (see doi: 10.1111/bph.13173.). 

Response: These are interesting findings. However, we currently do not have any evidence of increased oxidative stress in photoreceptor OS in the cKO mice, although our recent paper suggests that loss of XBP1 leads to premature age-related retinal degeneration associated with reduced glycolysis. We are currently generating new diabetic mice to investigate the metabolic changes as potential mechanisms for retinal degeneration in XBP1 cKO retinas. 

 The mitochondrial redox chain proteins as well as ATP synthase were shown to be ectopically expressed, not only in the mitochondria. In particular, this was found in the rod Outer Segments (see the data from Calzia, et al.): in 2019 Authors cannot ignore the data on the existence of an extra-mitochondrial oxidative phosphorylation in the rod OS (see Bruschi et al doi: 10.1021/acs.jproteome.7b00741.). Notably, an extra-mitochondrial oxidative phosphorylation was reported also in many cellular membranes (see for example the work from Mangiullo, et al., doi: 10.1016/j.bbabio.2008.08.003.; Arakaki et al., Mol Cancer Res. 2003 Nov;1(13):931-9; Shimizu N, et al. Am J Physiol Heart Circ Physiol. 2007 Sep;293(3):H1646-53

Response: We thank you for the reference. Please refer to the response to the prior comment.

Data from ERG show significant change only in the  a-wave of the electroretinogram,   which reflects the activity of photoreceptors, interstingly the only one  affected. Clearly the absence of effect on b-wave rules out a major role played by Mueller cells and inflamation, which imay instead be a consenquence of the oxidative stress primarily occurring in the rod OS due to hypèoxia and hyschemia due to impaiired blood supply, caused by the disease. . Undoubtedly,  the primary retinal damage occurs to photoreceptors, and rods in particular, and especially the outer segments (OS). In fact, oxidative stress is produced not only in the inner segment containing the mitochondria, but also in the Outer limb (see the cited work from Roelke et al. and Clazia et al, for example doi: 10.1016/j.biochi.2016.03.016.).

Response: We reported that only b-wave amplitudes show significant reduction in XBP1 cKO mice in both scotopic and photopic ERGs. We found there is no change in a-wave amplitudes.

I recommend that Authors reformulate the paper taking into consideration these emerging data, Authors must perform retinal immunohistochemistry utilizing and antibody against one of the subunits of the proteins of the respiratory chain or to FoF1-ATP synthase, in order to follow the morphology of the rod OS, evidencing at least morphologically its early impairment. Nice would also be the ability to test one of the respiratory complexes on retinal unfixed sections, but this thechnique may be longer to perform 

Also,  the mouse model of AMD obtained by stress induced by Blue light irradiation (Roelke et al. doi: (10.1371/journal.pone.0071570) must be cited.

Response: We thank you for the suggestions. As discussed in our earlier responses, we are currently generating new diabetic mice for measuring metabolic changes including mitochondrial respiratory function and glycolysis. This experiment will take another 6 – 8 months to complete. Furthermore, our previous data do not suggest a primary role of mitochondrial dysfunction in XBP1 deletion induced retinal degeneration (McLaughlin et al, Mol Neurodegen, 2018).

Round  2

Reviewer 1 Report

The manuscript by McLaughlin et al. is suitable for publication.

Minor comment:

1) Line 355: “diminished” and “affected”; one of the terms should be deleted.

Reviewer 3 Report

considering the Authors' studies are in progress and the experiments I have suggested they should do may not be within a suitable time frame for this Journal, 

I  accept their refusal to acknowlegde my concerns  and look forward to seeing their future results on oxidative stress in a model of DR , hoping  -for the sake of scientific progress- they consider the existence of a light-triggered source of oxidative stress within the rod OS,  related to phototransduction and bioenergetics.

On the other hand, I am fine about my main  concern, with the data in new Figure 4C.

as it is, the manuscript is quite "classically-oriented"  but formally and methodologically correct, therefore acceptable,